# Hemoglobin point-of-care testing in rural Gambia: Comparing accuracy of HemoCue and Aptus with an automated hematology analyzer

**Stefan A. Nass**[1,2]*, **Ilias Hossain**[2◉], **Chilel Sanyang**[2◉], **Bakary Baldeh**[2◉], **Dora I. A. Pereira**[2,3]

1 Medical Humanities, Amsterdam-UMC—VUmc Location, Vrije Universiteit, Amsterdam, The Netherlands, 2 Medical Research Council Unit, The Gambia at the London School of Hygiene & Tropical Medicine, Banjul, The Gambia, 3 Department of Pathology, University of Cambridge, Cambridge, United Kingdom

◉ These authors contributed equally to this work.
* stefan.a.nass@gmail.com

## Abstract

### Background

Anemia is one of the most impactful nutrient deficiencies in the world and disproportionately affects children in low-resource settings. Point-of-care devices (PoCDs) measuring blood hemoglobin (Hb) are widely used in such settings to screen for anemia due to their low cost, speed, and convenience. Here we present the first iteration of Aptus, a new PoCD which measures Hb and hematocrit (HCT).

### Aim

To evaluate the accuracy of Aptus and HemoCue® Hb 301 against an automated hematology analyzer (Medonic®) in Gambian children aged 6–35 months and the Aptus' usage in the field.

### Methods

Aptus, HemoCue® and Medonic® were compared using venous blood (n = 180), and Aptus and HemoCue® additionally using capillary blood (n = 506). Agreement was estimated using Bland-Altman analysis and Lin's concordance. Usage was assessed by error occurrence and user experience.

### Results

Mean Hb values in venous blood did not significantly differ between Aptus and HemoCue® (10.44±1.05 vs 10.56±0.93g/dl, p>0.05), but both measured higher Hb concentrations than Medonic® (9.75±0.99g/dl, p<0.0001). Lin's coefficient between Aptus and Medonic® was $r_c$ = 0.548, between HemoCue® and Medonic® $r_c$ = 0.636. Mean bias between the PoCDs venous measurements was -0.11g/dl with limits of agreement (LoA) -1.63 and 1.40g/dl. The

**Data Availability Statement:** All relevant data are within the manuscript and its Supporting Information files.

**Funding:** The IHAT-GUT study was supported by a Bill & Melinda Gates Foundation Grand Challenges New Interventions in Global Health award [OPP1140952]. The Nutrition Group of the MRC Unit The Gambia at LSHTM are supported by core funding MC-A760-5QX00 to the MRC Unit The Gambia/MRC International Nutrition Group by the UK MRC and the UK Department for the International Development (DFID) under the MRC/DFID Concordat agreement. Entia Ltd provided the Aptus devices and consumables and supported S. A.N with a travel award. The funders had no role in study design, data collection and analysis, decision to publish, or preparation of the manuscript.

**Competing interests:** The authors have declared that no competing interests exist. D.I.A.P. is one of the inventors of the IHAT iron supplementation technology, for which she could receive future awards to inventors through the MRC Awards to Inventor scheme. D.I.A.P. has served as a consultant for Vifor Pharma UK, Entia Ltd, Danone Nutritia, UN Food and Agriculture Organization (FAO) and Nemysis Ltd. D.I.A.P. has since moved to full employment with Vifor Pharma UK. S.A.N received a travel award from Entia Ltd. This does not alter our adherence to PLOS ONE policies on sharing data and materials.

bias was larger for the comparisons between the Medonic® and both Aptus (0.69g/dl, LoA 0.92 and 2.31g/dl) and HemoCue® (0.81g/dl, LoA 0.17 and 1.78g/dl). ROC curves showed an AUC of 0.933 in HemoCue® and 0.799 in Aptus. Capillary Hb was higher with Aptus than HemoCue® (10.33±1.11g/dl vs 10.01±1.07g/dl, p<0.0001). Mean bias was 0.32g/dl with LoA of -1.91 and 2.54g/dl. Aptus' usage proved intuitive, yet time-to-results and cuvettes could be improved.

## Conclusion

Both PoCDs showed a relatively limited bias but large LoA. Aptus and HemoCue® showed similar accuracy, while both overestimated Hb levels. Aptus showed promise, with its operation unimpaired by field conditions as well as being able to show HCT values.

## Introduction

Anemia is one of the most prevalent nutrient deficiencies worldwide. The most recent data from the World Health Organization (WHO) in 2011 estimate that the prevalence of anemia in children under the age of five is 42.6% worldwide, with the highest prevalence in the African region at 62.3% [1]. The most prominent causative factor is iron deficiency anemia (IDA), which has been shown to have detrimental immediate and long-term impact on growth, organ function and cognitive development in children [1–4]. IDA is estimated to affect 1.24 billion people worldwide, leading to 35 million disability-adjusted life years (DALYs) and contributing the largest share to total years lived with disability (YLD) [1, 5, 6]. The standard measurement to determine anemia status occurs via blood hemoglobin (Hb) levels, which enables quantitative but not causative assessment of anemia [1, 7]. As iron deficiency, defined in young children as inflammation-adjusted ferritin concentration < 12µg/L, is more costly and technologically sophisticated to measure, IDA is commonly assumed to represent 50% of anemia in a given area when lacking data on iron status [5]. However, anemia can be caused by other factors such as infections, other micronutrient deficiencies or hemoglobinopathies [5]. As such, implementing tests to screen for the cause of anemia can prevent misdiagnosis and consequently mistreatment. One approach is to measure the ratio of Hb to hematocrit (HCT), referred to as the mean corpuscular Hb concentration (MCHC), which decreases in microcytic anemias such as IDA [8, 9].

Furthermore, HCT is often used to approximate Hb in low-resource and field settings where direct Hb measurement is not possible or feasible. The correlation of HCT with Hb is, however, not sufficiently consistent [9]. Therefore, measuring HCT and Hb together would provide a more refined and robust procedure for anemia diagnostics suitable to low-resource settings. To this end, point-of-care devices (PoCDs) that are affordable, durable and easy to use are required. Presently, the most commonly used PoCD, set as the standard device in the guidelines for Demographic and Health Surveys (DHS) [10], is the HemoCue® series, which requires a simple fingerprick and provides results seen as comparable to automatic hematology analyzers [11, 12]. However, it solely measures Hb.

### Aptus device

The Aptus is a novel PoCD certified in accordance with the requirements of the EU Directive for *in vitro* diagnostics medical devices (directive 98/79/EC) and provides Hb and HCT

measurements simultaneously via centrifugation and photometry. More portable than the HemoCue® Hb 301, Aptus is a handheld device that could improve the in-field diagnosis of anemia in low- and middle-income countries (LMICs). The Aptus model used here was one of the first available commercially. Data from the NHS Healthcare Trust at Imperial College, London, showed good correlation with the Sysmex® bioanalyzer [13]. However, evidence to support its use in rural community-based studies in LMICs or with children is not yet available.

### HemoCue® device

The HemoCue® has been extensively evaluated and is seen as the standard PoCD for this type of setting [10]. However, as the results on its accuracy are mixed, its status as standard device is not undisputed [11, 14–19]. For HemoCue® Hb301, only one other study has been conducted in young children in rural low-resource settings [14].

### Aims

This paper presents the first comparative pilot study of these devices in LMIC field settings in rural clinics. The study aimed to: (1) compare Hb concentrations measured in venous blood by HemoCue® Hb 301 and Aptus with a gold standard automated hematology analyzer in young children with IDA in a rural and remote community in the Upper River Region of The Gambia; (2) compare the Hb measurements provided by HemoCue® Hb301 and Aptus in capillary blood; (3) determine the usability of Aptus in these settings and 4) highlight potential improvements to the Aptus model used here before a wider global launch. This was an ancillary study embedded within the IHAT-GUT clinical trial (Registration Number: NCT02941081) [20].

## Materials and methods

### Devices

The specifications of the analytical devices used in this study are summarized in Table 1. The HemoCue® Hb 301 and Aptus Model 10-100-01 used in this study are both PoCDs that can measure Hb in either capillary blood, collected from a finger prick, or venous blood, diluted with appropriate anticoagulants such as EDTA. The Aptus requires a smaller minimum sample volume of 4–8 μl (Table 1). The HemoCue® is able to measure also in arterial whole blood. The HemoCue® is larger and heavier than the Aptus, and both the cuvette storage and operating temperatures have a more limited range for HemoCue®. Humidity and altitude ranges are not quantified by the manufacturer of the HemoCue®. While the Aptus has an integrated rechargeable battery, the HemoCue® can be operated either by changeable batteries or with a power outlet. The HemoCue® has connectivity features allowing collected results to be read out on a computer, which this Aptus model lacked. Both devices have a built-in self-test and are calibrated to internationally recognized standards. Because Aptus measures Hb and HCT simultaneously, it requires an extra centrifugation step and has a measurement time of 60 seconds. The HemoCue® measures only Hb in 10 seconds. Both devices utilize photometry for their measurements: the Aptus measures three wavelengths and the HemoCue® two. Centrifugation requires the Aptus to be placed on a reasonably level surface that is not disturbed by significant movement during sample processing. The Aptus does not directly measure Hb, as is the case for the HemoCue®. Rather, it measures HCT and MCHC of the cellular constituents after centrifugation, and calculates Hb by multiplying MCHC with HCT. Additionally, the Aptus can display results in relation to pre-specified reference ranges. These

**Table 1. Specifications for HemoCue® Hb 301 and Aptus device [21–23].**

|  | *HemoCue® Hb 301* | *Aptus* |
|---|---|---|
| *Dimensions* | 140 × 70 × 160 mm | 78 x 83 x 52 mm |
| *Weight* | 500g (incl. batteries) | 96g |
| *Storage temp.* | Analyzer: 0–50˚C (32–122˚F) | Analyzer: 0–50˚C (32–122˚F) |
|  | Cuvettes: unopened 10–40˚C (50–104˚F); short-term storage (6 weeks) -18–50˚C (0–122˚F); 12-month open vial stability | Cuvettes: unopened or opened and resealed 0–50˚C (32–122˚F) |
| *Storage humidity* | - | 0–90% |
| *Operating temp.* | 10–40˚C (50–104˚F) | 5–45˚C (41–113˚F) |
| *Operating humidity* | - | 0–90% |
| *Altitude* | - | < 4500m |
| *Power* | AC adapter (6v) or 4 AA batteries | Micro USB (5V) or integrated Li-Ion battery |
| *Interface* | Printer and HemoCue® | Micro USB |
|  | Basic Connect™ including optional barcode scanner |  |
| *Quality Control* | Built-in self-test, optional liquid controls | Internal quality control. Self-Calibration and self-checks before every measurement, optional liquid controls |
| *Principle* | Absorbance measurement of whole blood at an Hb/HbO2 isosbestic point at 506 nm and 880 nm | Centrifugation and photometry at 515 nm, 660 nm and 940nm |
| *Calibration* | Factory calibrated against the ICSH method | Hb: Factory calibrated against HiCN method. HCT: Microhematocrit method |
| *Sample Material* | Capillary, venous (EDTA suitable), arterial whole blood | Capillary, venous (EDTA, Heparin suitable) |
| *Sample Volume* | ~10 µL | 4–8 µL |
| *Measurement Range* | 0–25.6 g/dL (0–256 g/L, 0–15.9 mmol/L) | Hb: 5–25 g/dl |
|  |  | HCT: 0.15–0.75 L/L (15–75%) |
| *Measurement Time* | < 10 seconds | 60 seconds |
| *Price* | Analyzer: ~£400 | Analyzer: £250 |
|  | Consumables: ~£80 per 200 tests | Consumables: £60 per 200 tests |

include the WHO and the Dacie and Lewis ranges, which can be corrected for altitude, sex, pregnancy status and age via the touchscreen [21–23]. The automated hematology analyzer used as the clinical laboratory gold standard was the Medonic® M-Series M16S/M20S Auto-loader (hereafter Medonic®).

To assess usability of the Aptus in field settings, a log of error messages as displayed by the Aptus was kept. These included overfilling or underfilling of cuvettes, termination of centrifugation due to tilt or vibrations of the device, and incorrect cuvette placement. These errors, combined with the user experience of the single operator of the Aptus in this study, were used to extrapolate possible improvements for the device in future model versions.

## Study setting and participants

This prospective study was conducted between 8 October and 22 November 2018 and was ancillary to the IHAT-GUT trial conducted in the Bassé government area, Upper River Region (URR), The Gambia [20]. The climate is Sahelian, following a pattern of rainy season from June-October and dry season from November-May [24]. During the study period, average temperature and humidity around Bassé were 31˚C with 67% humidity in October, then 29˚C with 58% humidity in November [25]. The Gambia ranks 174 out of 189 on the Human Development Index [26], with the rural URR representing the most socio-economically deprived region of the country, exhibiting limited access to health services or appropriate infrastructure.

This was particularly the case for the study area, located in the Wuli and Sandu districts approximately 400 km inland of the capitol Banjul, on the opposite, northern side of the river Gambia to Bassé, the only major town in the district [24]. This area is predominantly composed of subsistence farming communities from which the study participants were recruited. The Bassé region consistently scores below the national average on most morbidity and mortality measures, with the latest Demographic and Health Survey from 2011 (DHS) listing under-five mortality rates at 92 per 1000 (1 in 11) and the proportion of stunted children at 32% [24]. Furthermore, 82.5% of children in the Bassé area meet the WHO criteria for anemia of Hb< 11.0 g/dl, with 56.4% and 6.5% meeting the criteria for moderate and severe anemia, respectively [24].

All subjects were evaluated and screened for IHAT-GUT study eligibility. In the present study, this was a convenience sample of all children enrolled in the IHAT-GUT study and who attended the study clinics at the pre-specified study timepoints for blood collection. In short, participants were otherwise healthy children of 6–35 months with IDA at enrollment, defined as Hb between 7–11 g/dl and serum ferritin below 30 µg/l, from 17 villages in the study area. Exclusion criteria included severe malnutrition, severe anemia, congenital anomalies and diarrhea or dysentery. For a comprehensive list, please refer to the main study protocol [20].

## Ethical issues

Ethical approval for this ancillary study to the IHAT-GUT trial [20] was obtained from The Gambia Government/MRCG Joint Ethics Committee (SCC1632, 3/10/2018). No additional intervention or participant procedures were necessary for this study. As such, written informed consent of the parent was obtained for each participant as part of the IHAT-GUT main trial before the study procedures commenced. All staff participating in recruitment were trained in GCP, informed consent procedures and translation of the provided information material into local languages. In case of illiteracy, the trial was explained in understandable terms in a local language in the presence of an impartial witness in a room with appropriate privacy. In case of literacy, parents were given sufficient time to read the information material. Sufficient time was also given to address any questions parents might have. No child took part in the trial without prior informed consent, and all participants were protected according to the Sponsor Clinical Trial/Non-Negligent Harm Insurance and Medical Malpractice Insurance. For further information we refer to the IHAT-GUT protocol [20].

## Hemoglobin measurements

The IHAT-GUT study protocol defines that on study day 85, venous blood is collected instead of capillary blood. These samples were used in the present study for the comparisons between the PoCDs and the automated hematology analyzer. Venipuncture was performed by the IHAT-GUT study nurses following standardized study protocols. Briefly, the participant's arm was wiped with 70% ethanol. A 21-gauge Multifly® needle was used for venipuncture. The samples for the Medonic® analysis were collected in an EDTA Microtainer® blood collection tube, which was continuously inverted for 30 seconds immediately after collection, thus far exceeding the recommended minimum of 10 inversions. Immediately after the blood tubes were filled, the needle was removed and the blood that remained in the needle and attached tube systems was drained on a plastic surface to form a large drop, from which both Hemo-Cue® and Aptus samples were collected (gravity method). The EDTA tubes were subsequently transported to the laboratory site at the MRC field station in Bassé in an isolated carrying box containing cooling elements, where they were analyzed with the Medonic® within four hours of collection.

For all other study timepoints, participants visited the study clinics weekly for a finger prick performed by the nurse following standardized study protocol. These capillary blood samples were used for the comparisons between the two PoCDs, HemoCue® and Aptus. We note that the way the finger prick was performed as part of IHAT-GUT differed from what the Aptus manufacturer recommends, i.e. blood was collected from the thumb and slight pressure was applied for the blood drop to form. Briefly, the child's thumb was wiped with 70% ethanol and pricked with a Sherwood Safety blood lancet. A little pressure was applied to the finger so that a domed drop formed. Capillary blood for both HemoCue® and Aptus was collected from separate drops of blood from the same fingerprick to minimize variability due to the collection technique. The first drop was wiped clean with cotton, the second drop was used for the measurement with the HemoCue®, and the third drop was used for the Aptus measurement by one of two devices. HemoCue® measurements were conducted by one IHAT-GUT study nurse, whereas the Aptus was exclusively operated by the present study lead author. If during screening a measurement was Hb <7 g/dl (the WHO definition of severe anemia) a venous blood sample was taken and analyzed with Medonic within four hours. If this confirmed the field measurement, the child was classified as severely anemic and stopped the study supplementation to receive standard treatment with oral iron syrups, as per Gambian National Guidelines. Therefore, both PoCDs in this study were evaluated for their usefulness as screening tools, while the analyzer served as diagnostic standard. As such, the WHO standard as cut-off was valid for both PoCDs, which is in line with manufacturers' recommendations. Indeterminate screening results, i.e. an Hb ≥7g/dl, were thus not analyzed with the reference standard as per the IHAT-GUT protocol [20]. In the present study, both indeterminate test results and reference standards were used in the comparative analysis and handled as continuous variables and not categorized based on any Hb threshold.

HemoCue®, Aptus and Medonic® Hb measurements were all performed in accordance with the manufacturers' operating manuals. Results were recorded on the devices and then manually in a notebook. The study nurses also collected further clinical information during field studies as pertaining to the IHAT-GUT study. The nurses, lab staff and the principal investigator performing the Aptus tests were aware of Hb reference values via their biomedical training.

## Quality control (QC)

Before each measurement, the Aptus self-calibrates and self-checks during the centrifugation phase to account for environmental influences, such as temperature and humidity. The device is also able to check for underfilling and overfilling, can remove air bubbles by centrifugation, has an internal QC and displays error messages if it fails.

The HemoCue® has an internal quality control "self-test", which the system performs automatically every time the analyzer is turned on. The test is performed at regular intervals (typically every two hours) if the analyzer remains switched on. If the self-test fails, an error code will be displayed. Additionally, the Eurotrol Hb301 external QC (levels 1 and 2) was used to verify the accuracy of HemoCue® Hb301 devices daily.

For the Medonic®, the external QCs were run at the start of each analysis day. If the QC results were within range, then the study samples were tested. The laboratory supervisor approved the QC results daily. For external quality control, the laboratory participates in an external inter-laboratory proficiency scheme with OneWorld Canada for hematology full blood analysis and has good clinical laboratory practice (GCLP) accreditation by Qualogy Ltd (UK).

## Data analysis

Aptus displayed error messages in 7 out of 190 venous blood measurements (4%) and 108 out of 615 capillary blood measurements (17.6%) (Fig 1). These were due to under- or over-filling

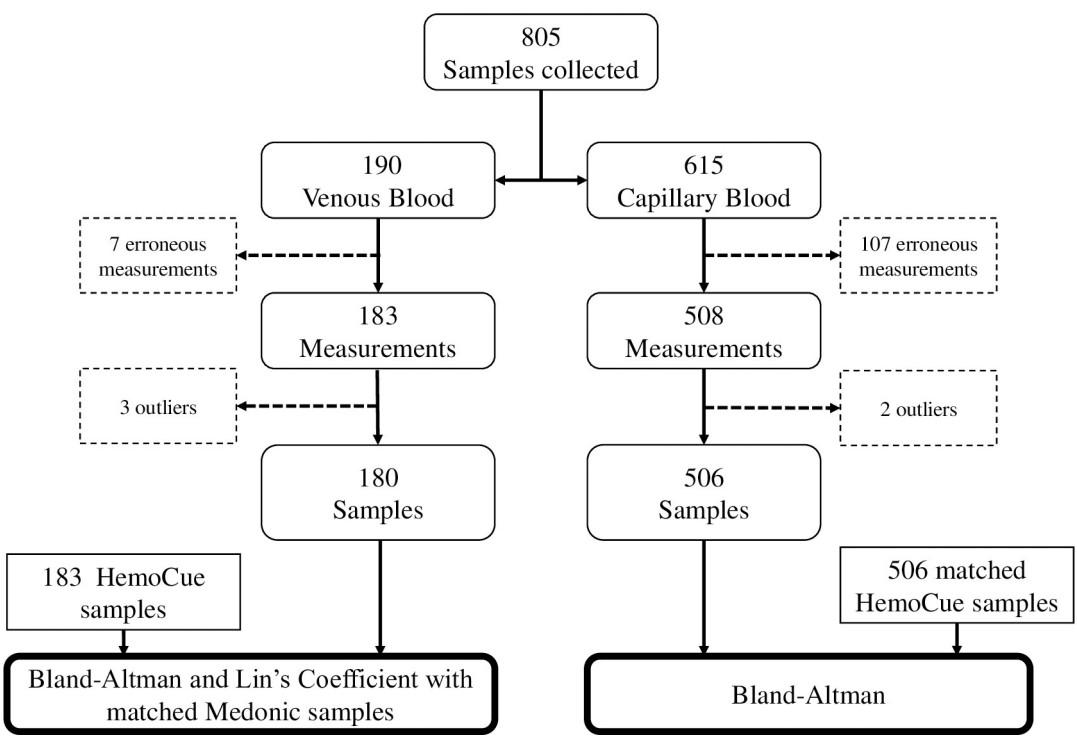

**Fig 1. Flowchart of study design.** Excluded samples (dashed rectangles), total of Aptus (rounded rectangles) and HemoCue®
(rectangles) data points used for the statistical analyses are indicated. For venous blood, both Aptus and HemoCue® samples
were taken during venipuncture and matched with the corresponding Medonic® measurement from the same sample tube.
For capillary blood, every Aptus data point was matched with the corresponding HemoCue® measurement.

of the cuvettes and are expected in such high throughput settings with inexperienced users.
HemoCue® did not display error messages during this study, irrespective of how well the
cuvettes were filled. Medonic®'s error measurements are repeated in the lab and, therefore,
no further sample measurements had to be excluded from the analysis presented here. Of the
691 valid sample measurements, 183 were collected from venous blood and 508 from capillary
blood (Fig 1).

As this study occurred between study day 57 and 113 of the IHAT-GUT trial, some partici-
pants had developed severe anemia since the beginning of the study (Tables 2 and 3) and were
subsequently discontinued from supplementation in the IHAT-GUT trial to receive standard
treatment. Since this study presents a methodological comparison of the devices, the clinical
consequences of the measurements do not pertain to our results and these low Hb values were
therefore not excluded from our analysis.

Sample size for IHAT-GUT was determined as per the published study protocol [20]. The
present study is an exploratory method comparison pilot study. As such, as there is no avail-
able performance data of Aptus in field settings, no data is available to perform a meaningful
power or sample size calculation.

SPSS Statistics version 25 (IBM corp.) was used for data analysis and visualization. Using
three times IQR as cutoff, three Aptus measurements (15.4g/dl, 18.5g/dl and 20.7g/dl) from
venous blood were considered outliers and removed from further analysis, leaving 180 venous
blood measurements for analysis (Fig 1). Two Aptus Hb measurements (15.9g/dl and 18.6g/dl)
in capillary blood were considered outliers, and 506 capillary blood measurements were used
for the final analysis (Fig 1). Normality was confirmed for the resulting datasets using the

Kolmogorov-Smirnov test. Paired t-tests and repeat measures ANOVA with the Greenhouse-Geisser correction and Bonferroni post-hoc analysis were used for comparisons between groups. Computation of Lin's Concordance Correlation Coefficient (abbreviated $r_c$) was conducted using a macro S1 Macro. Lin's coefficient analysis was conducted for the comparisons with the automated analyzer in venous blood as it measures both association and agreement. It represents a modification of Pearson's correlation coefficient, in that it assesses not merely the distance of data points to the line of best fit, but also how far this line deviates from the line of perfect agreement, as represented by the 45-degree line through the origin. Lin's coefficient has a range of -1 to 1, with $r_c = 1$ representing perfect agreement, in which all data points lie on the 45-degree line. As points either depart from this line, or as the line of best fit departs from the 45-degree line, the $r_c$ value diminishes [27, 28].

## Results

### Demographics

A total of 805 samples (190 venous blood and 615 capillary blood) were collected from 223 study participants enrolled in cohort 3 of the IHAT-GUT trial (Table 2). After excluding samples with erroneous measurements (see above) and data outliers, a total of 686 Aptus Hb measurements were included in the final dataset (Fig 1).

### Venous blood Hb concentration by Aptus and HemoCue® compared to the automated hematology analyzer

Each measurement included in this set of comparisons (Table 3) is from an individual child. The mean Hb concentrations obtained by the Aptus (10.4±1.0 g/dl) and HemoCue® (10.6 ±0.9 g/dl) devices were not statistically different (Table 3, p = 0.15). The mean difference in Hb between Aptus and HemoCue® Hb measurements in venous blood was -0.11 (95% CI: -0.23, 0.00) g/dl (Fig 2).

**Table 2. Baseline characteristics of study participants.**

| Age at enrollment (months) | |
|---|---|
| Mean (SD) | 19.58 (7.26) |
| Median (IQR) | 20 (11) |
| Range (min, max) | 6, 35 |
| **Sex** | |
| Female | n = 117 |
| Male | n = 106 |
| **Z–Scores** | |
| Mean Weight-for-Age (SD) | -0.84 (0.83) |
| Mean Height-for-Age (SD) | -0.69 (0.87) |
| Mean Weight-for-Height (SD) | -0.77 (0.82) |
| **Hb (g/dl) Medonic (n = 218)** | |
| Mean (SD) | 9.14 (0.91) |
| Median (IQR) | 9.15 (1.20) |
| Range (min, max) | 6.9, 11.2 |
| **Ferritin (µg/l) (n = 218)** | |
| Mean (SD) | 17.08 (21.99) |
| Median (IQR) | 11.75 (14.03) |
| Range (min, max) | 0.1, 247.5 |

**Table 3. Blood hemoglobin concentrations as determined by Aptus, HemoCue®, and Medonic® in venous blood samples.**

| | Aptus Hb (g/dl) | HemoCue® Hb (g/dl) | Medonic® Hb (g/dl) |
|---|---|---|---|
| **Number of datapoints** | 180 | 183 | 183 |
| **Minimum** | 7.7 | 7.5 | 6.7 |
| **25% Percentile** | 9.7 | 10.1 | 9.1 |
| **Median** | 10.4 | 10.6 | 9.8 |
| **75% Percentile** | 11.0 | 11.2 | 10.4 |
| **Maximum** | 13.2 | 13.5 | 13.2 |
| **Mean** | 10.44 | 10.56 | 9.75 |
| **Std. Deviation** | 1.05 | 0.92 | 0.99 |
| **Std. Error of Mean** | 0.08 | 0.07 | 0.07 |
| **Lower 95% CI** | 10.29 | 10.43 | 9.61 |
| **Upper 95% CI** | 10.60 | 10.69 | 9.90 |
| **Number of results $\geq$ 11 g/dl (non-anemic)** | 53 | 65 | 20 |

Three data outliers were removed for Aptus. PoCD measurements were conducted on blood drops taken from a venous blood sample before it was collected into the EDTA blood tube for the Medonic® measurements.

Both PoCDs measured significantly greater Hb concentrations than the Hb value obtained by the Medonic® automated hematology analyzer (Table 3, p<0.0001 for both comparisons). The mean difference between Hb concentration measured by Aptus vs Medonic® was 0.69 (95% CI: 0.57, 0.81) g/dl and that of HemoCue® vs Medonic® was 0.81 (95% CI: 0.73, 0.88) g/dl (Fig 2). In both cases more than 60% of measurements showed differences in Hb concentration between -1 and 1 g/dl (Fig 2).

Bland-Altman plots for the Hb concentration comparisons in venous blood samples are shown in Fig 3. There is very little bias between the two PoCDs for the measurement of Hb concentration with mean difference of -0.11 (95% CI: -0.228, 0.00) g/dl (Fig 3A). The Aptus measurements (Fig 3B) show slightly broader limits of agreement than the HemoCue® measurements (Fig 3C) for the comparison with the Medonic® automated hematology analyzer. The biases for Aptus of 0.69 (95% CI: 0.571, 0.813) g/dl (Fig 3B) and for HemoCue® of 0.81 (95% CI: 0.733, 0.879) g/dl (Fig 3C) compared to the Medonic® are similar.

The Desirable Biological Variation Database specifications propose a total error allowable (TEa) for Hb of ±4.19% from the true value as acceptable for a device measuring hemoglobin [29]. Taking the Medonic® measurements as the true value, the Aptus measured 122 measurements (68%) and the HemoCue® 159 measurements (88%) outside of this range.

Overall, if we consider the commonly used clinically meaningful difference in Hb concentrations of ± 1 g/dl (29), in the Aptus vs HemoCue® comparisons, 83.9% of values fell within this 1 g/dl threshold, for Aptus vs Medonic® this was 63.3% of values and for HemoCue® vs Medonic®, it was 69.4% of values.

Lin's coefficient analysis plots are shown in Fig 4. The correlation coefficient ($r_c$) for the comparison of Hb concentration determined by Aptus vs HemoCue® was $r_c$ = 0.688 (95% CI: 0.604, 0.756), for Aptus vs Medonic® $r_c$ = 0.548 (95% CI: 0.458, 0.627) and for HemoCue® vs Medonic® $r_c$ = 0.636 (95% CI: 0.571, 0.694). For all of these comparisons, there is a positive correlation between methods but the agreement is sub-optimal. Data for the comparison of hematocrit measurements in venous blood between Aptus and Medonic® are shown in S1 Table and S1 and S2 Figs.

A ROC analysis was also conducted on the venous blood comparison. As there was only one confirmed case of severe anemia and the median values across all devices were close to or

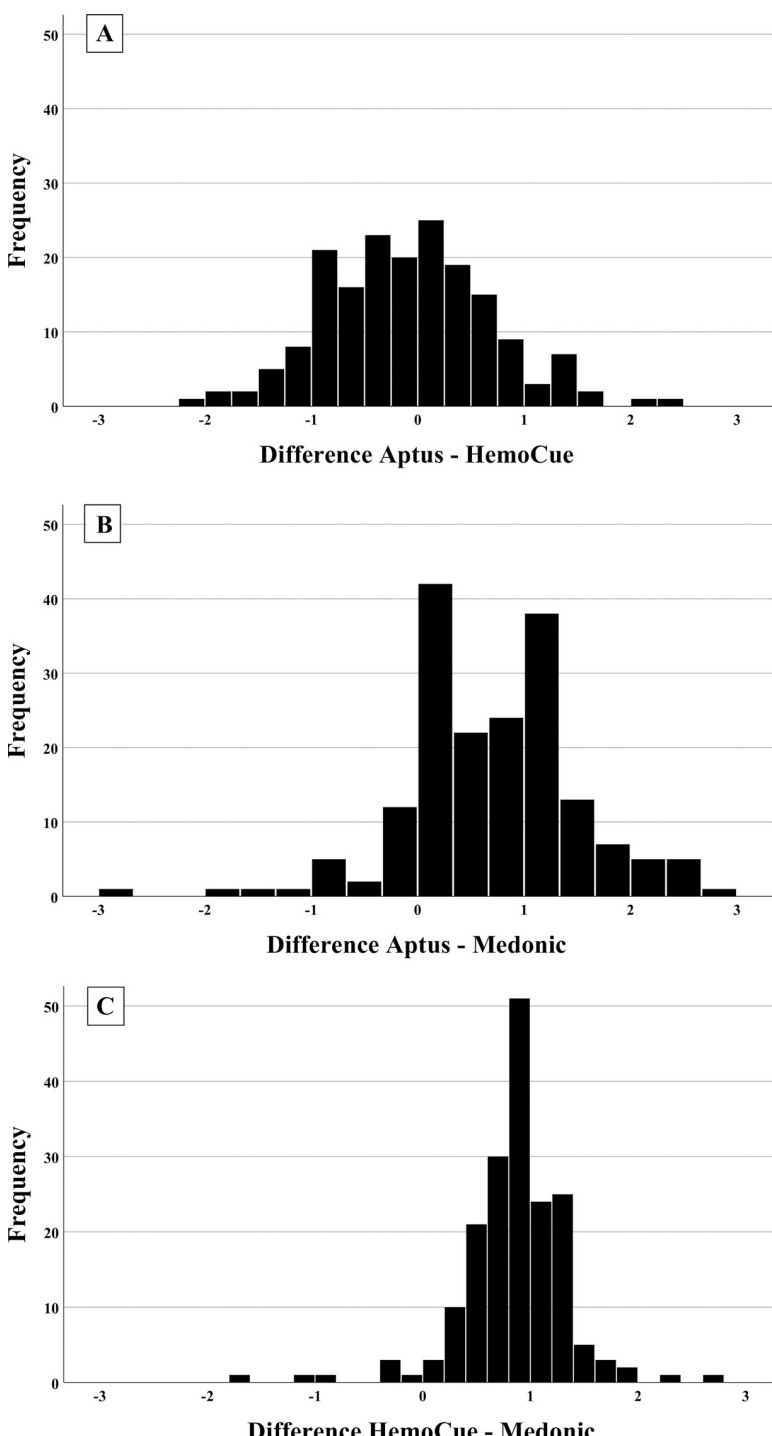

**Fig 2. Histograms of the differences in hemoglobin (Hb) concentration measured in venous blood samples.**
Aptus-HemoCue® (panel A) and Aptus-Medonic® (panel B) comparisons, n = 180; HemoCue®-Medonic® (panel C), n = 183. The frequency of each difference is represented as a % of total measurements. For these comparisons, 151 (83.9%) of values in panel 1, 114 (63.3%) in panel 2 and 127 (69.4%) in panel 3 showed differences in Hb concentration between -1 and 1 g/dl.

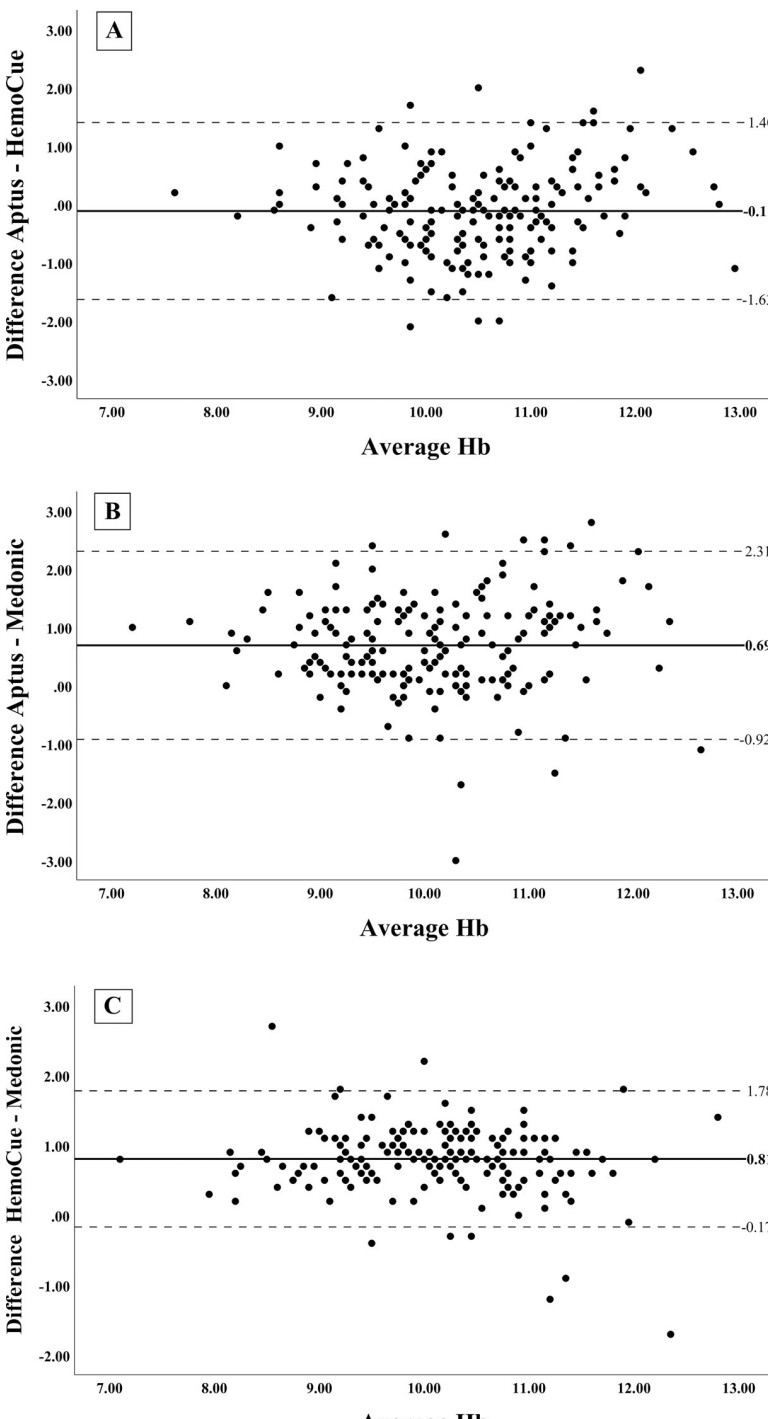

**Fig 3. Bland-Altman comparison of the hemoglobin measurements performed in venous blood samples.** Data represents the difference versus the average Hb results between Aptus-HemoCue® (panel A), Aptus-Medonic® (panel B) and HemoCue®-Medonic® (panel C). The solid lines represent the mean difference between the measurements (bias), while the dotted lines indicate the 95% limits of agreement between methods.

above the threshold for moderate anemia of 10 g/dl (Table 3), these were deemed unusable as an outcome measure. Instead, absence of anemia (Hb $\geq$ 11g/dl) was taken as a binary outcome

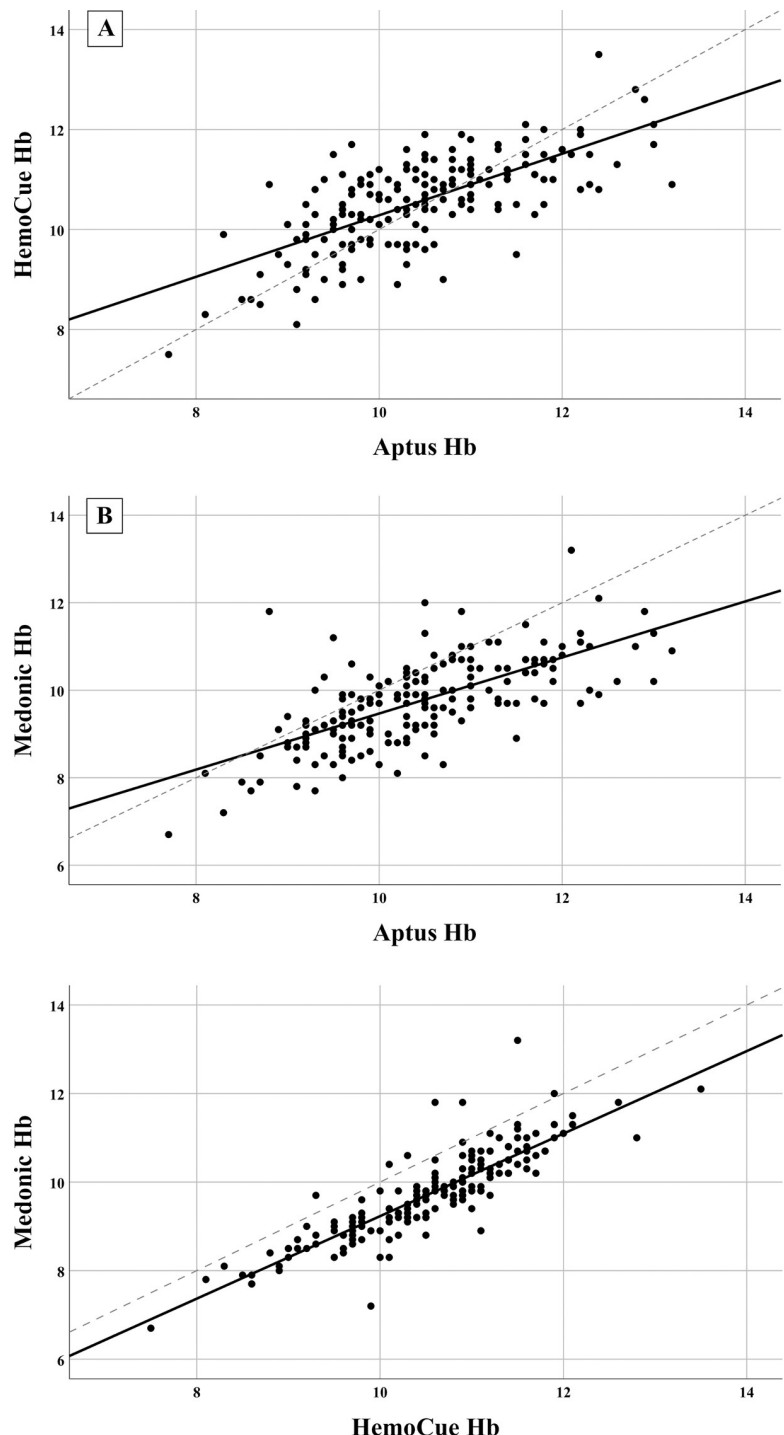

**Fig 4. Lin's Concordance Correlation Coefficient graphs for venous Hb comparison.** The dotted line represents the 45-degree line of perfect agreement through the origin, while the solid blue line is the line of best fit. As the Aptus-HemoCue® panel 1 (n = 180) and Aptus-Medonic® panel 2 (n = 180) indicate, the Aptus data has a more pronounced spread in both graphs. Furthermore, the line of best fit deviates significantly from the 45-degree line than in the HemoCue®-Medonic® comparison, indicating weaker agreement. Conversely, the spread of data and angle of the line of best fit in the HemoCue®-Medonic® comparison in panel 2 (n = 183) illustrate that agreement between both PoCDs is stronger compared to the Aptus.

measure to compare PoCDs with the lab analyzer, using the reference standard outcome as the true measure. As can be seen in Fig 5, HemoCue® was superior, with an AUC of 0.933 (Std. Error: 0.023, 95% CI: 0.878, 0.989) compared with the Aptus' AUC of 0.799 (Std. Error: 0.06, 95% CI: 0.681, 0.916).

## Hb concentration measured by two PoCDs, Aptus and HemoCue®, in capillary blood

A total of 506 measurements from 223 children were included in this set of comparisons. Mean Hb concentrations measured in capillary blood by Aptus (10.33±1.11g/dl) and HemoCue® (10.01±1.07g/dl) were statistically different (Table 4, p<0.0001). The mean difference between Aptus and HemoCue® Hb concentration was 0.32 (95% CI 0.22–0.42) g/dl (Table 4). There were 170 out of 506 (33.6%) measurements where the difference between the two measurements was above the ± 1g/dl clinically significant threshold.

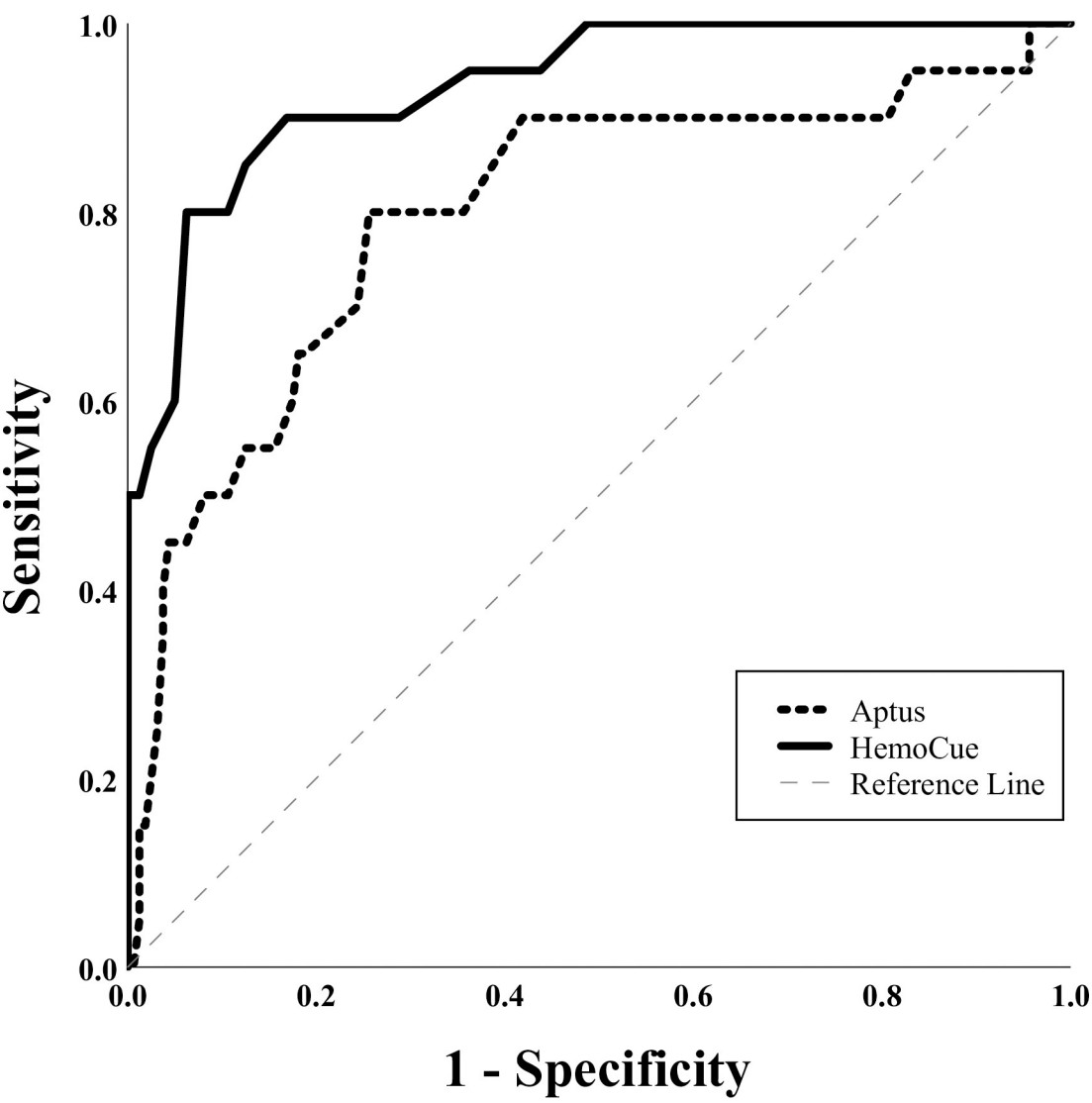

**Fig 5. ROC curve comparison of Aptus and HemoCue®.** HemoCue® (solid line) proved superior to Aptus (dashed line) in accuracy of measuring non-anemia, with an AUC of 0.933 compared to 0.799, respectively.

**Table 4. Blood hemoglobin concentrations as determined by Aptus and HemoCue® in capillary blood samples.**

|  | Aptus Hb (g/dl) | HemoCue® Hb (g/dl) | Aptus-HemoCue® Hb difference (g/dl) |
|---|---|---|---|
| Number of datapoints | 506 | 506 | 506 |
| Minimum | 6.4 | 7.3 | -3.1 |
| 25% Percentile | 9.5 | 9.2 | -0.4 |
| Median | 10.4 | 10.0 | 0.3 |
| 75% Percentile | 11 | 10.8 | 1.0 |
| Maximum | 13.6 | 13.4 | 4.5 |
| Mean | 10.33 | 10.01 | 0.32 |
| Std. Deviation | 1.11 | 1.07 | 1.13 |
| Std. Error of Mean | 0.05 | 0.05 | 0.05 |
| Lower 95% CI | 10.23 | 9.92 | 0.22 |
| Upper 95% CI | 10.43 | 10.11 | 0.42 |

Two data outliers for the Aptus were removed. Capillary blood was collected from the same finger prick but measurements were conducted in two independent blood drops.

The Bland-Altman plot for the comparison between the two PoCDs is shown in Fig 6. There is no significant relationship between the bias and the average Hb result. The distance between the limits of agreement (-1.91g/dl, 2.54g/dl) is comparatively wide at 4.45 g/dl.

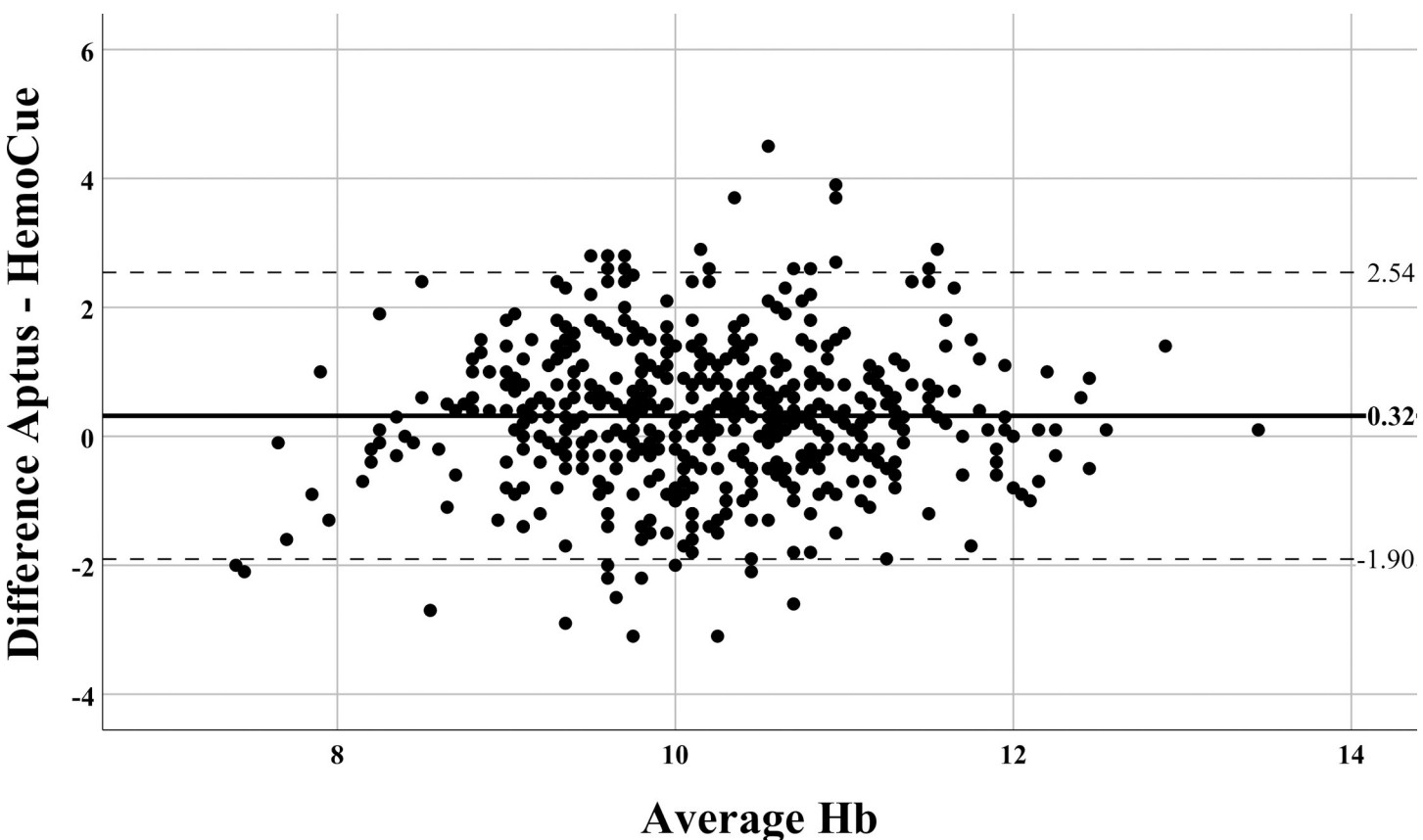

**Fig 6. Bland-Altman comparison of the hemoglobin measurements performed in capillary blood samples by Aptus and HemoCue®.** Data represents the difference (Aptus–HemoCue®) versus the average Hb results. The solid lines represent the mean difference between the measurements (bias), while the dotted lines indicate the 95% limits of agreement between methods.

## Aptus device usage

The Aptus device proved easily portable and intuitive in use. The two Aptus devices used in this study were sufficient to cover the maximum of 85 participants per day. However, as photometric measurement with Aptus only commences after centrifugation has completed, the time-to-result is significantly longer in Aptus than HemoCue® (Table 1). This could be a drawback in very high-throughput settings when there is not much turnaround time between patients being screened. Additionally, the centrifugation process is sensitive to both significant vibrations and spatial orientation, which led to instances in which a measurement was not initiated or aborted. This did not pose a major impediment to normal usage when the manufacturer's recommendations were followed. Moreover, errors in sampling are only detected with photometry, which in the majority of cases made it impossible to collect a second sample from the same participant, as they had already been processed by the IHAT-GUT nurses and left the clinic. Errors in capillary blood sampling most commonly occurred due to underfilling of the cuvette. The manufacturer's instructions recommend that the capillary blood should form a dome shaped droplet of sufficient size before attempting to fill the cuvette. However, many children were distressed by the fingerprick, making it difficult to produce a droplet and appropriately fill the Aptus cuvette with the resulting blood film formed on the finger. This would probably be improved when collecting blood in adults, with more experienced users or improved cuvettes. This is in contrast to the HemoCue® cuvette, which draws blood on contact in any position and the device does not indicate errors from under- or over-filling. Contrarily, when placing a drop of venous blood from the needle on a plastic surface, the most common error with the Aptus was overfilling.

## Adverse events

Besides the observation that blood drawing caused some discomfort to the children, the study reported no adverse events related to blood collection (capillary or venous). Abnormal blood results, i.e. severe anemia (Hb<7 g/dl) were identified in 0 samples with HemoCue® and 3 samples with Aptus. As the Aptus comparison was ancillary to the IHAT-GUT study, its results did not trigger a subsequent analysis for severe anemia by Medonic® as would have been the case if severe anemia was identified by HemoCue®. However, during this study there was one identified severe anemia case with Medonic® measurements collected as part of the outcome assessment timepoints. As per IHAT-GUT protocol this participant was discontinued from the IHAT-GUT supplementation and received standard treatment.

## Discussion

The large impact and preventable nature of anemia warrant large scale screen-and-treat efforts, and cost-efficient and accurate PoCDs are a crucial element in this. The Aptus represents a new development by measuring Hb and HCT concentration. This study assessed the accuracy and usability of the new Aptus PoCD and the well-established HemoCue® 301 PoCD against the automated hematology analyzer Medonic® M-series, in an ancillary study to the IHAT-GUT clinical trial set in a community setting in rural The Gambia [20]. To our knowledge, this study is the first to test the accuracy of Aptus in a low-income rural setting in children.

Overall, both PoCDs showed similar levels of agreement with the laboratory analyzer, however Aptus showed slightly lower bias and wider limits of agreement than the HemoCue®. Both PoCDs were shown to overestimate Hb concentration, as has been shown in previous studies with HemoCue® in children in similar settings [14, 30].

Venous blood measurements exhibited better agreement both between the PoCDs and with the laboratory analyzer in comparison to the agreement observed between the PoCDs in

capillary blood. This is in line with previous research that shows PoCD agreement to be inferior in capillary sampling [7, 14, 30], which is unsurprising given the mixed venous/arterial nature of capillary blood [31], within-subject variation between drops from the same finger prick [32] and dependency on proper technique for consistency of results [33]. This is reflected in the present study by wider limits of agreement in the Bland-Altman plots, the difference of which far exceeds the clinically acceptable inaccuracy of ± 1g/dl. This indicates that the within-subject variation of measurements is large and the methods are thus not unequivocal in a clinically relevant manner. Furthermore, SD values were larger for the capillary samples despite the much larger sample size than in the venous comparison, thereby also indicating more variability of results. The method used to obtain a PoCD sample from the venous blood by placing a drop on a surface first (gravity method), might be of influence here, as it was found to be more accurate than collection from finger prick in a study on the HemoCue® in Rwanda [30]. While this has not been previously evaluated for Aptus, it could partly explain the superior results in venous sampling for both PoCDs. However, whereas both devices are capable of measuring Hb in venous blood, their principal function is to be used as a rapid test in capillary blood collected from a fingerprick, particularly in low-resource field settings where PoCDs largely function as a primary diagnostic tool. Therefore, although direct comparison between the PoCDs and the laboratory analyzer from one venous sample clearly minimized biases, and is thus the most suitable method for comparing accuracy, this study also included the capillary blood comparison between the two PoCDs as this better represents their intended use.

The overestimation of Hb concentration by both PoCDs could translate to underestimating anemia rates when these devices are used as the only diagnosis tool. Averaging multiple capillary measurements, either by using multiple cuvettes [34] or collecting multiple drops of blood in a tube and sampling from it [32], have been proposed to eliminate variation and improve accuracy. However, such measures require both more time and materials, thereby making them less feasible in LMICs and high-throughput screenings in particular. Alternative non-invasive devices have not yet been shown to be consistently reliable [6, 35, 36], so more advanced PoCD capabilities—such as the MCHC measurement of the Aptus—would improve the value of invasive procedures. This holds for low-resource settings in particular, where full blood count analysis is often unavailable and PoCDs are the modus operandi.

Thus, usability and appropriateness to such settings are important considerations. Whereas the HemoCue® series has long been recommended as standard for field settings in low resource areas due to its portability, cost and user-friendliness [14], its performance is not without criticism as studies on it provide mixed results [7, 14, 15, 18, 30, 37–39]. Reasons for this variability are thought to include, among others, climatological factors, handling and storage of the cuvettes, and sampling technique [11, 15, 40]. Furthermore, the lack of a specified working range for both altitude and humidity is suboptimal, as both have been shown to influence reliability [39, 40]. Regardless of these constraints, the usage of the HemoCue® in this study proved simple, rapid, and without noteworthy malfunctions.

To our knowledge, this is the first instance in which this Aptus model was tested in high-throughput and low-resource community settings in children. Aptus is far smaller in dimension and lighter than the HemoCue®, making it more portable, and has validated wide ranges of altitude, temperature and humidity. A major advantage of the Aptus is that it measures MCHC and HCT directly. As MCHC has intrinsic diagnostic value, it would be beneficial to have the option to display MCHC directly post-measurement. This is particularly relevant since this study revealed poorer agreement in the HCT values than for the calculated Hb values (Figs S2 and 4, respectively), which is interesting as the Aptus uses the gold standard microhematocrit method of determining hematocrit by centrifugation [41] whereas the flow-cytometry based models like the Medonic® calculate HCT. This suggests that the MCHC has a

strong corrective influence in the Hb calculation, and is thus an important variable to show the device's accuracy. Future studies should investigate this further.

The usage proved intuitive and simple, yet some limitations apply specifically to these settings, namely the longer duration of measurement and difficulty in handling the cuvettes. If the intended use is high-throughput screenings in rural field clinics, then the comparatively longer measurement time due to centrifugation may be a disadvantage in comparison to HemoCue®. A further consideration is that the battery of the Aptus model used was not replaceable and the lack of a consistent power supply for recharging batteries in this type of setting could lead to issues with continuous usage.

Based on field experience and statistical analysis, the Aptus appears comparable in measurement capabilities to the standard of the HemoCue® given the limitations cited. It is stressed that the Aptus results may have been confounded by not adhering strictly to the manufacturer's specifications. This included the frequent inability to obtain a large dome-shaped blood droplet in the young children, use of the thumb for finger pricking instead of the second or third digit, and squeezing of the finger in order to procure a sufficient amount of blood from the child. This blood sampling procedure is discouraged since the resulting expulsion of interstitial fluid can skew the measurement [33]. However, as the HemoCue® sample was taken using the same technique, a resulting bias can be assumed to be present in both measurements. As the HemoCue® and Aptus samples were always taken from the second and third drop of blood produced, respectively, this bias would be comparable within measurements per device, while it is still possible that the composition between second and third drop may have varied due to this technique. Furthermore, the larger number of error messages observed with Aptus was related to the lower level of experience with this device than with HemoCue®, which we note as a limitation. Furthermore, the Aptus was operated by only one user, which could have introduced bias, whereas HemoCue® sampling was conducted by one of the five study nurses who all had been using the device for ten months prior to this study, through which possible biases would have been balanced and sampling technique could have been superior due to increased experience with the device.

Overall, to improve usability of Aptus in high throughput settings by a wide range of users, experienced or inexperienced, we recommend improvements to both the cuvettes and the timing of the error messages in the device, and the incorporation of additional battery charging capabilities. We note that the Aptus model used in this study is one of the first available commercially and, as a result of this study and feedback from other users, has undergone further development to improve its accuracy, software interface and cuvette handling.

## Conclusion

This was the first study of the Aptus device in this type of setting and was intended to inform further development in terms of performance before wider adoption in resource-poor rural areas. Aptus and HemoCue® showed similar accuracy and both overestimated Hb levels, which could lead to an underestimation of anemia prevalence when used as a screening or diagnostic tool. Aptus showed promise with its operation unimpaired by the field clinic's conditions, but operational limitations concerning duration of measurements may make it less suitable for high-throughput screening. Importantly, the additional diagnostic value of measuring MCHC may make Aptus a superior choice if the improvements highlighted in this study are achieved. Since MCHC analysis was beyond the scope of this paper, future studies should investigate this. By broadening the range of validated PoCDs available to users, we hope that anemia diagnosis and screening programs improve coverage for the benefit of billions of people around the world.

## Supporting information

**S1 Data.**
(XLSX)

**S1 Macro. SPSS macro for Lin's Concordance Correlation Coefficient.** Macro used in the study to calculate the Lin's coefficient in SPSS. Available from: https://gjyp.nl/marta/Lin.sps. Date accessed: 23.04.2019.
(DOCX)

**S1 Table. Comparison of hematocrit (HCT) measurements between Aptus and Medonic®.** Measurements in the Aptus were conducted in blood drops taken from a venous blood sample. Al HCT values given as percentage. The mean difference between the methods is 5.11 (95% CI: 4.76, 5.45) %, which was statistically significant (p<0.0001).
(DOCX)

**S1 Fig. Bland-Altman plot for hematocrit (HCT) measured in venous blood by Aptus and Medonic®.** Data represents the difference (Aptus–Medonic®) versus the average HCT results. The solid lines represent the mean difference between the measurements (bias) is 5.1%, while the dotted lines indicate the 95% limits of agreement between methods of 0.5 (lower) and 9.71% (upper) limit.
(TIF)

**S2 Fig. Lin's Concordance Correlation Coefficient for hematocrit (HCT) measured in venous blood by Aptus and Medonic®.** The dotted line represents the 45-degree line of perfect agreement through the origin, while the solid blue line is the line of best fit. Lin's coefficient is $r_c = 0.285$, and there is significant divergence from the 45-degree line and spread around the line of best fit (S2 Fig). It must be noted that the Aptus® does not provide decimals for HCT while the Medonic® does. This explains the columnar nature of the data in both plots, and negatively influences agreement due to an inherently lower accuracy of the Aptus®'s output.
(TIF)

## Acknowledgments

We would like to sincerely thank all participants and their families for their participation and the local communities for welcoming us to conduct this study. We would like to thank Entia Ltd for supplying the Aptus devices and consumables, for user training, and for reviewing the final draft of this manuscript.

## Author Contributions

**Conceptualization:** Stefan A. Nass, Dora I. A. Pereira.

**Data curation:** Stefan A. Nass.

**Formal analysis:** Stefan A. Nass.

**Investigation:** Stefan A. Nass, Ilias Hossain, Chilel Sanyang, Bakary Baldeh.

**Methodology:** Stefan A. Nass, Dora I. A. Pereira.

**Project administration:** Stefan A. Nass, Ilias Hossain, Dora I. A. Pereira.

**Supervision:** Dora I. A. Pereira.

**Validation:** Dora I. A. Pereira.

**Visualization:** Stefan A. Nass.

**Writing – original draft:** Stefan A. Nass.

**Writing – review & editing:** Stefan A. Nass, Ilias Hossain, Chilel Sanyang, Bakary Baldeh, Dora I. A. Pereira.

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
