## [Decision Letter · Decision Letter 0]

19 May 2020

PONE-D-20-03905

Hemoglobin point-of-care testing in rural Gambia: Comparison of HemoCue® and Aptus™ with an automated hematology analyzer

PLOS ONE

Dear Mr. Nass,

Thank you for submitting your manuscript to PLOS ONE. After careful consideration, we feel that it has merit but does not fully meet PLOS ONE’s publication criteria as it currently stands. Therefore, we invite you to submit a revised version of the manuscript that addresses the points raised during the review process.

We would appreciate receiving your revised manuscript by Jul 03 2020 11:59PM. To enhance the reproducibility of your results, we recommend that if applicable you deposit your laboratory protocols in protocols.io, where a protocol can be assigned its own identifier (DOI) such that it can be cited independently in the future. For instructions see: http://journals.plos.org/plosone/s/submission-guidelines#loc-laboratory-protocols

We look forward to receiving your revised manuscript.

Kind regards,

Simeon-Pierre Choukem

Academic Editor

PLOS ONE

Journal Requirements:

'The authors have declared that no competing interests exist.'

We note that you received funding from a commercial source: Entia Ltd.

Reviewers' comments:

Reviewer's Responses to Questions

**Comments to the Author**

1. Is the manuscript technically sound, and do the data support the conclusions?

Reviewer #1: Partly

Reviewer #2: Yes

2. Has the statistical analysis been performed appropriately and rigorously? 

Reviewer #1: Yes

Reviewer #2: Yes

3. Have the authors made all data underlying the findings in their manuscript fully available?

Reviewer #1: Yes

Reviewer #2: Yes

4. Is the manuscript presented in an intelligible fashion and written in standard English?

Reviewer #1: Yes

Reviewer #2: Yes

5. Review Comments to the Author

Reviewer #1: This study aimed to (1) compare Hb concentrations measured in venous blood by 2 PoCDs with an automated analyzer in young children with IDA in a rural setting (2) compare the Hb measurements provided by 2 PoCDs in capillary blood; (3) determine the usability of Aptus in these settings and 4) highlight potential improvements to the Aptus model used here before a wider global launch.

1) The method used for assessing the last two objectives are not well described in article under the "Materials and Method" section

2) The abstract does not include the last 2 aims in it's objectives, nor the results but seems to jump to a conclusion on the use of one PoCD

3) Statistical analysis: It is mentioned in the paper that "The Clinical Laboratory Improvement Amendments (CLIA) of 1988 postulate a ±7% deviance from the true value as acceptable for a device measuring hemoglobin." I think the limit of 7% is very large for hemoglobin measurements. I wish to suggest to the authors to use the total error allowable (TEa) based on Desirable Biological Variation Database specifications. For Hb, this is 4.19%.(https://www.westgard.com/biodatabase1.htm#de). This would be more fair method for comparing.

4) I wish to suggest the addition of ROC curves for the 2 PoCDs versus Lab measurements in addition to the agreement analysis

5) The manuscript should follow the Standards for Reporting of Diagnostic Accuracy studies - STARD statement (http://www.stard-statement.org/) for properly reporting this type of study. A STARD checklist should ideally accompany this manuscript.

6) I think the Conclusion on the usability of Aptus at the end of this manuscript is not fully supported by the results of this study.

7) This study included children. Please kindly specify whether consent was obtained from parents/legal guardians before study procedures

Reviewer #2: The authors of this work present an interesting work which evaluates the performance of 2 Point-of-care devices (PoCDs), Aptus® and HemoCue® against an automated haematology analyser in children in a country with limited resources, The Gambia. These 2 devices showed similar level of agreement with the laboratory analyser but Aptus® showed slightly lower bias and wider limits of agreement than the HemoCue®. Both PoCDs were shown to overestimate Hb concentration. Then, Aptus® could represent an alternative to HemoCue® as standard for field settings in low resource areas.

The study design is clear and the results well presented. I agree with the conclusions and recommendations of this work. However, some clarifications will help to improve the quality of this work.

Introduction

The introduction is clear but would benefit from being more concise.

Table 1 (line 104): would it be possible to have an idea on the prices of these PoCDs, in order to be in agreement with the comment presented on line 369?

Materials and methods

Why the choice of 35 months as the upper limit for the recruitment of children, and not 36 months, even 59 months as in the WHO report (The Global Prevalence of Anaemia in 2011)?

Children with severe anaemia were excluded from this study (line 124). For children between 6 and 59 months of age, WHO defines severe anaemia as <7.0 g / dL. Can you explain why in the basic characteristics of the participants in your study, the minimum haemoglobin level is 6.9 g / dL? (Table 2). Same as in Table 3 where the minimum Hb level for Medonic is 6.7 g / dL?

Results

Do you have climate data (particularly the temperature) at the time the blood samples were taken and the analysis of these samples was performed?

Discussion

The discussion is clear but would benefit from being more concise.

6. PLOS authors have the option to publish the peer review history of their article (what does this mean?). If published, this will include your full peer review and any attached files.

Reviewer #1: Yes: Colette Sih

Reviewer #2: Yes: Fernando KEMTA LEKPA

---

## [Author Response · Author response to Decision Letter 0]

26 Jun 2020

Dear editors,

We would like to thank the reviewers for their work and have addressed their concerns in the manner described below.

Reviewer #1: 

This study aimed to (1) compare Hb concentrations measured in venous blood by 2 PoCDs with an automated analyzer in young children with IDA in a rural setting (2) compare the Hb measurements provided by 2 PoCDs in capillary blood; (3) determine the usability of Aptus in these settings and 4) highlight potential improvements to the Aptus model used here before a wider global launch.

1) The method used for assessing the last two objectives are not well described in article under the "Materials and Method" section

Clarification has been added in lines 98 – 102

2) The abstract does not include the last 2 aims in it's objectives, nor the results but seems to jump to a conclusion on the use of one PoCD

Descriptions have been added to the abstract (lines 8, 11, 20,21,23-24)

3) Statistical analysis: It is mentioned in the paper that "The Clinical Laboratory Improvement Amendments (CLIA) of 1988 postulate a ±7% deviance from the true value as acceptable for a device measuring hemoglobin." I think the limit of 7% is very large for hemoglobin measurements. I wish to suggest to the authors to use the total error allowable (TEa) based on Desirable Biological Variation Database specifications. For Hb, this is 4.19%.(https://www.westgard.com/biodatabase1.htm#de). This would be more fair method for comparing.

The CLIA method has been replaced with the proposed TEa method (lines 292-295)

4) I wish to suggest the addition of ROC curves for the 2 PoCDs versus Lab measurements in addition to the agreement analysis

A ROC analysis was added with absence of anemia (Hb ≥ 11g/dl) as binary outcome variable (lines 316 – 326). As explained in the text, severe and moderate anemia were not suitable for comparison due to the low prevalence.

5) The manuscript should follow the Standards for Reporting of Diagnostic Accuracy studies - STARD statement (http://www.stard-statement.org/) for properly reporting this type of study. A STARD checklist should ideally accompany this manuscript.

Aspects of the text have been adapted to follow STARD guidelines and the checklist has been uploaded separately. We stress, however, that this was an exploratory comparative study and as such the intention was not to exhaustively determine the diagnostic accuracy of the PoCDs. Furthermore, the ancillary nature of this study necessitated adherence to IHAT-GUT study protocol, which was not planned as a comparative study between the devices and as such did not always represent the optimal study design for such a study. 

6) I think the Conclusion on the usability of Aptus at the end of this manuscript is not fully supported by the results of this study.

In order to address this, the usage experience with the Aptus device has been moved from the Discussion to the Results section (lines 353-373), and the Discussion adapted to more clearly highlight the combination of statistical analysis and user experience that our conclusion is based on.

 7) This study included children. Please kindly specify whether consent was obtained from parents/legal guardians before study procedures

Consent details have been specified (lines 132-143)

Reviewer #2: 

The authors of this work present an interesting work which evaluates the performance of 2 Point-of-care devices (PoCDs), Aptus® and HemoCue® against an automated haematology analyser in children in a country with limited resources, The Gambia. These 2 devices showed similar level of agreement with the laboratory analyser but Aptus® showed slightly lower bias and wider limits of agreement than the HemoCue®. Both PoCDs were shown to overestimate Hb concentration. Then, Aptus® could represent an alternative to HemoCue® as standard for field settings in low resource areas. The study design is clear and the results well presented. I agree with the conclusions and recommendations of this work. However, some clarifications will help to improve the quality of this work.

Introduction

1) The introduction is clear but would benefit from being more concise.

Parts of the Introduction have been reformulated in a more concise way, see tracked changes.

2) Table 1 (line 104): would it be possible to have an idea on the prices of these PoCDs, in order to be in agreement with the comment presented on line 369?

Prices have been added to the last row of table 1 (line 104).

Materials and methods

3) Why the choice of 35 months as the upper limit for the recruitment of children, and not 36 months, even 59 months as in the WHO report (The Global Prevalence of Anaemia in 2011)?

Use of 35 months cut-off (inclusive, we could also write 6 to <36 months) is because this inclusion criteria was for children under the age of 3 years old which has been the group used in the iron supplementation trials conducted in sub-Saharan Africa showing a potential detrimental effect of iron supplementation in infection and diarrhea. We could have used the under 5s as defined by WHO (up to 59 months) but the trials with iron have been focusing on under 35months. Furthermore, after 3 years the microbiome changes too close to an adult-like microbiome and impact of iron on microbiome was an important endpoint of IHAT-GUT and the hypothesis behind IHAT. You can reference:

https://pubmed.ncbi.nlm.nih.gov/16413877/

and

https://www.sciencedirect.com/science/article/pii/S0966842X19302148

4) Children with severe anaemia were excluded from this study (line 124). For children between 6 and 59 months of age, WHO defines severe anaemia as <7.0 g / dL. Can you explain why in the basic characteristics of the participants in your study, the minimum haemoglobin level is 6.9 g / dL? (Table 2). Same as in Table 3 where the minimum Hb level for Medonic is 6.7 g / dL?

Children had Hb ≥7 g/dL measured by the Medonic at the time of screening and enrolment in the study. The data presented in this manuscript was collected from study day 57 onwards, at which point some children had shown a drop of Hb to below the severe anemia cut-off. These children were subsequently discontinued of the study supplementation, as per IHAT-GUT study protocol, and given standard of care iron drops according to national guidelines in The Gambia. As this clinical consequence does not interfere with the methodological comparison of the measuring devices, these measurements were not excluded in the present study. 

For clarification, this explanation was added to the Methods section (lines 170-179, 217-222)

Results

5) Do you have climate data (particularly the temperature) at the time the blood samples were taken and the analysis of these samples was performed?

Temperature and humidity averages added (line 110,111).

Discussion

6) The discussion is clear but would benefit from being more concise.

Parts of the Discussion have been reformulated in a more concise way, see tracked changes. Furthermore, the description of the Aptus in-field usage has been moved to the Results section (lines 353-373), thereby significantly shortening the Discussion.

Academic Reviewer

1. PLOS ONE's style requirements, including those for file naming.

File names have been amended, Panels in figures 2-4 combined into planes, line numbers added to the title page and minor formatting issues fixed in the main text.

2. Competing interests and funding statements

Competing interests and funding statements have been amended and clarified as per the feedback received.

3. Figures

All figure files have been checked with Preflight Analysis and Conversion Engine (PACE) digital diagnostic tool. 

We believe the manuscript is now suitable for publication in your journal.

Kind regards,

Stefan Nass

On behalf of all the authors

---

## [Decision Letter · Decision Letter 1]

28 Jul 2020

PONE-D-20-03905R1

Hemoglobin point-of-care testing in rural Gambia: Comparing accuracy of HemoCue® and Aptus™ with an automated hematology analyzer

PLOS ONE

Dear Dr. Nass,

Thank you for submitting your manuscript to PLOS ONE. After careful consideration, we feel that it has merit but does not fully meet PLOS ONE’s publication criteria as it currently stands. Therefore, we invite you to submit a revised version of the manuscript that addresses the points raised during the review process.

ACADEMIC EDITOR:

The manuscript will be accepted once you have carried out the minor revisions required by reviewer 1

We look forward to receiving your revised manuscript.

Kind regards,

Simeon-Pierre Choukem

Academic Editor

PLOS ONE

Reviewers' comments:

Reviewer's Responses to Questions

**Comments to the Author**

1. If the authors have adequately addressed your comments raised in a previous round of review and you feel that this manuscript is now acceptable for publication, you may indicate that here to bypass the “Comments to the Author” section, enter your conflict of interest statement in the “Confidential to Editor” section, and submit your "Accept" recommendation.

Reviewer #1: All comments have been addressed

Reviewer #2: All comments have been addressed

2. Is the manuscript technically sound, and do the data support the conclusions?

Reviewer #1: Yes

Reviewer #2: Yes

3. Has the statistical analysis been performed appropriately and rigorously? 

Reviewer #1: Yes

Reviewer #2: Yes

4. Have the authors made all data underlying the findings in their manuscript fully available?

Reviewer #1: No

Reviewer #2: Yes

5. Is the manuscript presented in an intelligible fashion and written in standard English?

Reviewer #1: Yes

Reviewer #2: Yes

6. Review Comments to the Author

Reviewer #1: The aim of this study was to (1) compare Hb concentrations measured in venous blood by 2 PoCDs with an

automated analyzer in young children with IDA in a rural setting (2) compare the Hb

measurements provided by 2 PoCDs in capillary blood; (3) determine the usability of Aptus in these

settings and 4) highlight potential improvements to the Aptus model used here before a wider

global launch.

1) A considerable amount of grammatical errors noted with additional "the", "a", sentences which are too long making their comprehension difficult

Reviewer #2: Dear Editor,

We have read the new version of the article submitted for our review by Nass S. et al on “ Hemoglobin point-of-care testing in rural Gambia: Comparing accuracy of HemoCue® and Aptus™ with an automated hematology analyzer”.

The authors have taken all of my comments into consideration.

We have no new comments.

Best regards

fkl

7. PLOS authors have the option to publish the peer review history of their article (what does this mean?). If published, this will include your full peer review and any attached files.

Reviewer #1: **Yes: **Colette Sih

Reviewer #2: **Yes: **Fernando Kemta Lekpa

---

## [Author Response · Author response to Decision Letter 1]

28 Aug 2020

Dear editors,

We would like to thank the reviewers for their work and have addressed their concerns in the manner described below.

Reviewer #1: 

1) A considerable amount of grammatical errors noted with additional "the", "a", sentences which are too long making their comprehension difficult

The manuscript has been reviewed again by three native speakers. For the revisions, see tracked changes.

4. Have the authors made all data underlying the findings in their manuscript fully available? Answer: No

The reviewer has not further specified which data they are referring to. All data pertaining to this analysis is presented in the manuscript. The demographic and endpoint data for the IHAT-GUT trial will be published separately and the full database for the IHAT-GUT trial will be made available upon publication of the primary and secondary endpoints. We trust that this is sufficient with regards to the data policy.

We believe the manuscript is now suitable for publication in your journal.

Kind regards,

Stefan Nass

On behalf of all the authors

---

## [Editor Report · Decision Letter 2]

16 Sep 2020

Hemoglobin point-of-care testing in rural Gambia: Comparing accuracy of HemoCue® and Aptus™ with an automated hematology analyzer

PONE-D-20-03905R2

Dear Dr. Nass,

We’re pleased to inform you that your manuscript has been judged scientifically suitable for publication and will be formally accepted for publication once it meets all outstanding technical requirements.

Kind regards,

Simeon-Pierre Choukem

Academic Editor

PLOS ONE
---

## [Editor Report · Acceptance letter]

21 Sep 2020

PONE-D-20-03905R2 

Hemoglobin point-of-care testing in rural Gambia: Comparing accuracy of HemoCue and Aptus with an automated hematology analyzer 

Dear Dr. Nass:

I'm pleased to inform you that your manuscript has been deemed suitable for publication in PLOS ONE. Congratulations! Your manuscript is now with our production department. 

Kind regards, 

on behalf of

Dr. Simeon-Pierre Choukem 

Academic Editor

PLOS ONE